# Learning Diffeomorphic and Modality-invariant Registration using B-splines

**Huaqi Qiu**[1]                                                    HUAQI.QIU15@IMPERIAL.AC.UK
[1] *BioMedIA Group, Imperial College London, London, UK*
**Chen Qin** [1,2]                                                      CHEN.QIN@ED.AC.UK
[2] *Institute for Digital Communications, University of Edinburgh, Edinburgh, UK*
**Andreas Schuh**[1]                                            ANDREAS.SCHUH@IMPERIAL.AC.UK
**Kerstin Hammernik**[1,3]                                        K.HAMMERNIK@IMPERIAL.AC.UK
**Daniel Rueckert**[1,3]                                          D.RUECKERT@IMPERIAL.AC.UK
[3] *Technical University Munich, Munich, Germany*

## Abstract

We present a deep learning (DL) registration framework for fast mono-modal and multi-modal image registration using differentiable mutual information and diffeomorphic B-spline free-form deformation (FFD). Deep learning registration has been shown to achieve competitive accuracy and significant speedups from traditional iterative registration methods. In this paper, we propose to use a B-spline FFD parameterisation of Stationary Velocity Field (SVF) to in DL registration in order to achieve smooth diffeomorphic deformation while being computationally-efficient. In contrast to most DL registration methods which use intensity similarity metrics that assume linear intensity relationship, we apply a differentiable variant of a classic similarity metric, mutual information, to achieve robust mono-modal and multi-modal registration. We carefully evaluated our proposed framework on mono- and multi-modal registration using 3D brain MR images and 2D cardiac MR images.

## 1. Introduction

Image registration is an essential task in medical image analysis. Given a moving image $\mathcal{M}$ and a fixed image $\mathcal{F}$, image registration aims to find the spatial transformation $\phi$ which maps a location $\mathbf{x}$ in $\mathcal{F}$ to the location with corresponding tissue or structure in $\mathcal{M}$. In traditional approaches, a regularised transformation is embedded in an optimisation problem that minimises a dis-similarity metric between the fixed image $\mathcal{F}$ and the transformed moving image $\mathcal{M} \circ \phi$. This optimisation problem is commonly solved in an iterative way (Sotiras et al., 2013). Despite being effective, many iterations are often required to register one pair of images in such optimisation-based methods which can be very time consuming.

Recently, researchers have explored data-driven methods and the use of deep learning in image registration (Rueckert and Schnabel, 2020). Although often time-consuming in training, DL-based registration networks can perform one-pass registration during inference substantially faster than iterative optimisation methods. Recently proposed so-called *unsupervised* DL registration methods train networks using intensity-based similarity metrics instead of (synthesised) ground truth transformations (Balakrishnan et al., 2019; de Vos et al., 2019; Qin et al., 2018, 2020; Qiu et al., 2019). Most *unsupervised* methods use Mean Squared Error (MSE) or Cross-correlation (CC) as image matching criterion, and thereby

assume either identity or a strong linear relationship between the intensities of the images. However, these metrics are often ineffective when there are complex non-linear intensity relationships, e.g. in contrast-enhanced images or images from different modalities.

In this paper, we adopt a classic information-theoretic similarity metric, mutual information (MI), in *unsupervised* DL registration to handle non-linear intensity relationships. MI has been widely used in traditional registration algorithms for robust multi-modal registration (Pluim et al., 2003). However, MI is commonly computed using non-differentiable intensity histogram construction, making it challenging to be directly applied in DL registration. To compute MI in a differentiable way, we adopt the formulations introduced in Thévenaz and Unser (2000) and use continuous Parzen windows (PW) to estimate differentiable intensity distributions. In addition, we propose to use a B-spline FFD based diffeomorphic transformation model in our DL registration framework. Specifically, we use CNNs to learn a B-spline model of stationary velocity fields (SVF) over the entire image domain to obtain diffeomorphic transformations, to take advantage of the parameter-efficiency and intrinsic smoothness of B-splines.

The main contributions of our work are as follows: 1) We propose to learn a diffeomorphic SVFs parameterised efficiently by B-spline FFD to achieve fast and smooth diffeomorphic registration; 2) We use a differential formulation of mutual information in a whole-image DL registration framework to register images across modalities; 3) We carefully evaluate the introduced components on both mono-modal and multi-modal registration tasks using 3D brain MR images and 2D cardiac MR images.

## 2. Related works

Some recent learning-based registration methods specifically address multi-modal or modality-invariant registration problem. One approach is to use segmentation of the anatomical structures to guide registration, when a large amount of segmentation is available for training (Hu et al., 2018). Another approach is to reduce the problem to mono-modal registration via image-to-image translation (Arar et al., 2020) or disentanglement (Qin et al., 2019). These methods utilise powerful advances in deep learning generative models but cannot explicitly guarantee that the structures are not changed during the intensity transformation, and often have complicated frameworks that are non-trivial to train. Towards modality-invariant registration, Hoffmann et al. (2020) proposed to use contrast-varying synthetic images to train contrast-invariant registration networks. Most related to our work, de Vos et al. (2020) also adopts mutual information for DL registration. In contrast to their method, our approach uses fully convolutional network to parameterise registration over the entire image domain instead of patches and employs a diffeomorphic transformation model. We also evaluated our framework on more challenging inter-subject multi-modal registration tasks.

## 3. Methods

### 3.1. Mutual information in DL registration

Mutual information relaxes the linear intensity relationship constraint and measures information that one image contains about another image based on their intensity distributions. Studholme et al. (1999) later introduced Normalised Mutual Information (NMI) which is

more invariant to the amount of overlap between the two images. Here, we introduce the formulation of a differentiable estimate of NMI to enable its use in DL registration. In the context of image registration, NMI can be written as:

$$I_{NMI}(\mathcal{F}, \mathcal{M} \circ \phi) = \frac{H(\mathcal{F}) + H(\mathcal{M} \circ \phi)}{H(\mathcal{F}, \mathcal{M} \circ \phi)}, \tag{1}$$

where $H(\mathcal{F})$ and $H(\mathcal{M} \circ \phi)$ denote the marginal entropies of the fixed and moved images, and $H(\mathcal{F}, \mathcal{M} \circ \phi)$ denotes the joint entropy. The entropy of an image $\mathcal{I}$ can be defined as $H(\mathcal{I}) = - \int_i p(i) \ln(p(i)) di$ where $p(i)$ is the intensity distribution of image $\mathcal{I}$. To estimate this intensity distribution in digital images, one can construct histograms as discretised estimation of the distributions. This is usually achieved by counting the number of intensities or (intensity-pairs) that fall into intensity *bins*, which is mathematically equivalent to adding a rectangular window function centred at the intensity value of each point in the images to the histogram (pair of points for joint histogram). However, the rectangular window function makes the constructed histogram non-differentiable. To use MI in deep learning registration, we need a differentiable way to construct the intensity histogram to allow back-propagation and gradient-based training of the networks. To this end, we use a differentiable Parzen window (PW) (Thévenaz and Unser, 2000) instead of the rectangular window, as illustrated in Figure A2 in the Appendix. Formally, the joint histogram is computed as:

$$h(f, m) = \sum_{\mathbf{x} \in \Omega^\dagger} w \left( \mathcal{F}(\mathbf{x}) - f \right) w \left( \mathcal{M}(\mathbf{x} \circ \phi) - m \right), \tag{2}$$

where $f, m$ denotes intensity values on the fixed and moved images, $\Omega^\dagger$ denotes all points in the overlapping image domain, and $w(\cdot)$ is the Parzen window function. Normalising the joint histogram yields the joint distribution:

$$p(f, m) = \frac{h(f, m)}{\sum_{f \in L_\mathcal{F}, m \in L_{\mathcal{M} \circ \phi}} h(f, m)}. \tag{3}$$

where $L_\mathcal{F}$ and $L_{\mathcal{M} \circ \phi}$ denotes the *bin* centres where the histogram is evaluated. We opt to use the Gaussian function as the Parzen window, which fulfils the partition-of-unity constraint and is easy to compute, namely $w(i) = \frac{1}{\sqrt{2\pi}\sigma} \cdot \exp(-\frac{i^2}{2\sigma^2})$. The $\sigma$ is chosen so that the Full Width at Half Maximum (FWHM) of the function is one *bin*-width. The marginal distributions are estimated by marginalising the joint distribution, i.e. $p(f) = \sum_{m \in L_\mathcal{M}} p(f, m)$ and $p(m) = \sum_{f \in L_\mathcal{F}} p(f, m)$. Finally, we can compute the entropies and the NMI using Eq 1 with the joint and marginal distributions. We compute this effeciently using vectorised operations to combine with DL registration.

## 3.2. Registration framework

Here we introduce our deep learning registration framework, as illustrated in Figure 1. We focus on deformable registration of images after affine alignment. We follow the approach of (Dalca et al., 2018) and use a convolutional neural network (CNN) with parameters $\theta$ to map the fixed and moving images to the parameters of the transformation. Instead of directly outputting the velocity fields as in Dalca et al. (2018), our network (detailed in 3.2.2)

outputs the velocities of the B-spline control points $\boldsymbol{v}^{\beta}$, from which we can compute the transformation $\phi$ via B-spline tensor product and Squaring and Scaling (detailed in 3.2.1). During training, the moving image $\mathcal{M}$ is warped using $\phi$ via linear interpolation to acquire the moved image $\mathcal{M} \circ \phi$ which is then used to compute similarity loss $\mathcal{L}_{Sim}(\mathcal{F}, \mathcal{M} \circ \phi)$. We iterate over pairs of fixed-moving images in a training dataset to find the network parameters $\hat{\theta}$ that minimises the similarity loss $\mathcal{L}_{Sim}$ with the constraint of the regularisation loss $\mathcal{L}_{Reg}$. The overall loss can be written as,

$$\mathcal{L}(\mathcal{F}, \mathcal{M} \circ \phi) = \mathcal{L}_{Sim}(\mathcal{F}, \mathcal{M} \circ \phi) + \lambda \mathcal{L}_{Reg} \qquad (4)$$

where $\mathcal{L}_{Sim} = -I_{NMI}$ is the negative Normalised Mutual Information and $\mathcal{L}_{Reg}$ is a regularisation on the velocity field $\boldsymbol{v}$ to further enforce smoothness and diffeomorphism (Beg et al., 2005):

$$\mathcal{L}_{Reg} = \frac{1}{|\Omega|} \sum_{\mathbf{x} \in \Omega} \sum_{d \in D} \left\| \frac{\partial \boldsymbol{v}(x)}{\partial d} \right\|_2^2 \qquad (5)$$

where $\Omega$ denotes all points in the image domain and $d$ denotes the spatial dimension.

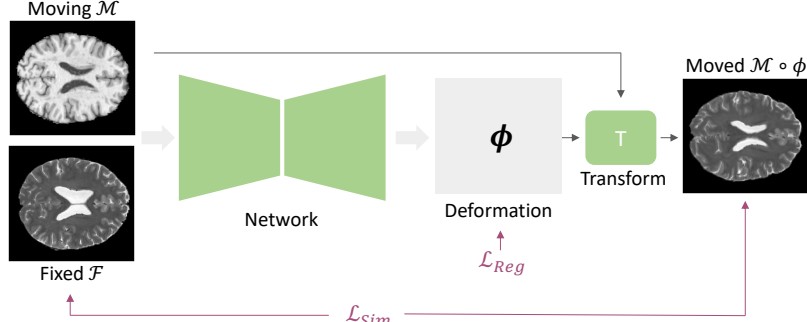

Figure 1: Our DL registration framework. The CNN predicts time-stationary velocities of the control points, and transformation is obtained by evaluating B-spline functions at the control points and efficient integration via Scaling and Squaring. The similarity loss based on MI and the smoothness regularisation loss is only needed for training.

### 3.2.1. TRANSFORMATION

Diffeomorphic transformation is topology preserving and invertible, which are desirable properties for some medical image registration applications. To ensure these properties, we use the flow of diffeomorphisms generated by the group exponential of spatially smooth Stationary Velocity Fields (SVFs). The diffeomorphic transformation $\phi$ is the group exponential of the time-stationary velocity field $\boldsymbol{v}$, i.e. $\phi = \exp(\boldsymbol{v})$, which can be efficiently computed using the Scaling and Squaring (SS) algorithm (Arsigny et al., 2006). To represent SVFs in a parameter efficient way while taking advantage of implicit smoothness of spline functions (Modat et al., 2012), we propose to use the cubic B-spline parameterisation of SVFs (SVFFD) in our deep learning registration framework. The CNN outputs the velocities of a grid of B-spline control points $\boldsymbol{v}^{\beta}$ with regular spacing $\boldsymbol{\delta}$. And the dense

velocity field is obtained using a weighted combination of cubic B-spline basis functions $\beta(\cdot)$ (Rueckert et al., 1999):

$$\boldsymbol{v}(\mathbf{x}) = \sum_{c \in C} \boldsymbol{v}_c^{\beta} \prod_{d \in D} \beta_d(\mathbf{x}_d - \boldsymbol{k}_{c,d})), \tag{6}$$

where $c$ is the index of the control points on the control point grid $C$, $\boldsymbol{k}$ denotes the coordinates of the control points in image space. The displacement field is obtained from the SVF via scaling and squaring. Since the B-spline basis function has limited local support, Eq. 6 can be implemented using transposed convolution with pre-computed B-spline basis functions as kernels.

### 3.2.2. NETWORKS

To learn the velocities of the B-spline control points $\boldsymbol{v}^{\beta}$, we use a fully convolutional network adapted from a U-net based architecture, as shown in Figure B3 in the Appendix. In order to support different control point spacings, we dynamically adapt the base U-net architecture so the output matches the size of the control point grid. To achieve this, we keep the U-net *decoder* layers that produce the largest feature map smaller than the size of the control point grid. Then we apply a linear interpolation layer to resize the feature map to match the size of the control point grid. This allows the use of arbitrary control point spacing. Finally, we apply three convolution layers to predict the output velocities. All convolution layers use a kernel size of 3 in all spatial dimensions and a LeakyReLU nonlinearity with negative slope of 0.2. No nonlinearity function is applied to the final layer. The same architecture is used for 2D and 3D.

## 4. Experimental Settings

### 4.1. Tasks and Data

We evaluate the proposed framework on three tasks: inter-subject 3D brain MRI registration of 1) T1w-T1w volumes; 2) T1w-T2w volumes; and 3) cardiac motion estimation via 2D registration between end-diastolic (ED) frame and end-systolic (ES) frame of cardiac MR images. For the brain registration task, we use 3D T1w and T2w images of 310 randomly selected subjects from the Cambridge Centre for Ageing and Neuroscience (CamCAN) project (Shafto et al., 2014; Taylor et al., 2017). The images have isotropic spatial resolution with voxel size of $1mm^3$ and are cropped to the size of $176 \times 192 \times 176$. All images are spatially normalised to a common MNI space using affine registration, skull-stripped using ROBEX[1] and bias-field corrected using the N4 algorithm in SimpleITK[2]. For evaluation, we also acquired the segmentation of 138 cortical and sub-cortical structures (grouped into 5 groups) automatically using MALPEM (Ledig et al., 2015). For cardiac motion estimation, we use 2D cardiac MR images of 210 subjects from the UK Biobank study[3]. The images have in-plane resolution of $1.8mm \times 1.8mm$ and are cropped to the size of $192 \times 192$. The segmentation of left ventricle cavity (LV), myocardium (MYO) and right ventricle (RV) are

---

1. https://www.nitrc.org/projects/robex
2. https://simpleitk.org
3. UK Biobank Imaging Study. http://imaging.ukbiobank.ac.uk

acquired using a state-of-the-art CNN-based automatic segmentation algorithm (Bai et al., 2018).

## 4.2. Evaluation metrics

We evaluate both the *accuracy* and the transformation *regularity* of registration. We evaluate the *accuracy* by measuring the overlap between the anatomical segmentation of the fixed image and the segmentation transformed by $\phi$ in the moving image using Dice score. The transformation *regularity* is evaluated based on the determinant of the Jacobian $J = |\nabla\phi|$. We evaluate the amount of points in the image that are "folded" due to the transformation by the ratio of points with $J < 0$. We also evaluate the spatial smoothness of the transformation by measuring the magnitudes of the gradient of the Jacobian determinant $|\nabla J|$.

## 4.3. Baseline comparisons

The presented deep learning registration framework using mutual information and B-spline SVF (denoted by "MIDIR") is firstly compared to a traditional iterative registration method based on the SVFFD transformation model. We also compare to a state-of-the-art DL registration method proposed in Dalca et al. (2018) (the deterministic version) combined with our differentiable mutual information, denoted by "VM$_{NMI}$". For the mono-modal tasks, we also compare to variants of methods that use localised normalised cross-correlation (LNCC) as similarity. The evaluation metrics at the initial affine registration ("Affine") is also provided for reference. The B-spline control point spacing $\boldsymbol{\delta}$ and regularisation weighting $\lambda$ are hyper-parameters. For all competing models, we carefully tuned these hyper-parameters using a held-out validation dataset while considering the balance between registration accuracy and transformation regularity. The hyper-parameters that yield best mean Dice score, with under 0.5% of the points with $|\nabla\phi| < 0$ were chosen. Hyper-parameter values of all results are shown in the Appendix C.

## 4.4. Implementation details

The traditional SVFFD method is implemented using Medical Image Registration ToolKit (MIRTK) (Schuh et al., 2014)[4]. The DL registration frameworks are implemented using Pytorch v1.5.1[5]. To reduce GPU memory usage, we compute NMI on a subset (50%) of randomly sampled positions in the image space at each iteration. The Adam optimiser was used with an initial learning rate of 1e-4. Learning rate decay of 1/10 per 50 epoch was used for all brain registration models. Running speed were measured on a workstation with an Intel® i7-8700 CPU and NVIDIA® Titan Xp GPUs. Our code is available at this url.

## 5. Results

Table 1 presents the quantitative evaluation of all models on all brain MR and cardiac MR registration tasks. We performed two-sided Wilcoxon signed-rank test to check for

---

4. https://mirtk.github.io

5. https://pytorch.org

statistical significance in differences between methods (significant if p-value is smaller than 0.05). On T1w-T1w tasks, the traditional SVFFD method outperformed all DL methods on Dice score while achieving good transformation regularity. Methods using LNCC as similarity achieved better Dice accuracy than the ones use NMI. Our MIDIR performed similarly to the baseline VM framework when the same similarity metric is used. On T1w-T2w registration, similar to T1w-T1w, methods using LNCC as similarity quantitatively achieved better results than those using NMI as similarity, with the traditional SVFFD using LNCC outperforming all other methods. The proposed MIDIR frameworks achieved similar Dice accuracy but significantly better regularity than the VM baseline. On the cardiac motion task, the DL models are on-par with traditional SVFFD when LNCC is used as similarity but significantly more accurate when NMI is used as similarity. Our MIDIR models achieved competitive accuracy and marginally better regularity on this task. The boxplots in Figure C4-C9 in the Appendix show the distribution of the results over test subjects. It can be noticed that traditional SVFFD produced more dispersed results with more outliers than DL methods. This could be an advantage of using a DL frameworks for the same transformation model, since the more complex CNN can learn a prior from a dataset to produce more consistent results. Some visual examples of the registration results are shown in Figure D10-D12 in the Appendix.

The runtime for each method to register one pair of 2D or 3D images are also reported in Table 1. Both CPU and GPU inference times are shown for a fair comparison with the CPU-based MIRTK (SVFFD). DL models are substantially faster on CPU and GPU. Our parameter-efficient MIDIR models run faster than the dense VM models especially on 3D tasks.

Table 1: Quantitative results on brain and cardiac registration tasks. The Dice score of different anatomical structures are averaged. $J_{<0}\%$ denotes the percentage of points with negative Jacobian determinant, higher means more "folding". $|\nabla_J|$ denotes the gradient magnitude of the Jacobian determinant, lower value means spatially smoother.

| | Brain T1w-T1w | | | Brain T1w-T2w | | | Cardiac Motion | | | Runtime (2D/3D) | |
|---|---|---|---|---|---|---|---|---|---|---|---|
| Methods | Dice | $J_{<0}\%$ | $|\nabla_J|$ | Dice | $J_{<0}\%$ | $|\nabla_J|$ | Dice | $J_{<0}\%$ | $|\nabla_J|$ | CPU | GPU |
| Affine | 0.619 | - | - | 0.619 | - | - | 0.500 | - | - | - | - |
| SVFFD$_{LNCC}$ | **0.836** | 0.107 | 0.024 | **0.770** | 0.150 | **0.027** | 0.781 | 0.161 | 0.045 | 43.3s/44min24s | - |
| VM$_{LNCC}$ | 0.814 | 0.295 | 0.051 | 0.753 | 0.176 | 0.047 | 0.797 | 0.094 | 0.034 | 115ms/17.7s | 6.48ms/228ms |
| MIDIR$_{LNCC}$ | 0.816 | 0.238 | 0.044 | 0.743 | 0.090 | 0.039 | **0.806** | 0.096 | 0.029 | 116ms/11.8s | 4.78ms/124ms |
| SVFFD$_{NMI}$ | 0.822 | 0.118 | **0.023** | 0.728 | 0.135 | **0.027** | 0.701 | **0.080** | **0.016** | 1min10s/3min34s | - |
| VM$_{NMI}$ | 0.807 | **0.106** | 0.038 | 0.733 | 0.197 | 0.047 | 0.797 | 0.151 | 0.036 | 115ms/17.7s | 6.48ms/228ms |
| MIDIR$_{NMI}$ | 0.813 | 0.121 | 0.038 | 0.735 | **0.023** | 0.028 | 0.803 | 0.151 | 0.033 | 116ms/11.8s | 4.78ms/124ms |

## 6. Discussion

The quantitative results show that the traditional SVFFD method outperforms the DL methods in several settings. SVFFD performs slow but detailed optimisation for each pair of images, while the DL methods perform one-pass fast predictions but less accurately in its current form. Our traditional baseline also employs a multi-resolution framework

which improves optimisation and registration accuracy, while the DL methods only use one resolution.

Our experiments also show that LNCC outperformed NMI on several settings, and perhaps interestingly so on T1w-T2w registration. The intensity relationships between T1w and T2w images in our dataset, when observed locally, can be roughly described by intensity inversion and could be handled by LNCC. However, (L)NCC can not be expected to be applicable to other multi-modal registration with more complex appearance and intensity relationships, such as MR-PET or MR-CT, while (N)MI is more generally applicable (Sotiras et al., 2013). Experiments with more multi-modal data is therefore required to demonstrate this, which we will investigate in the future. On the other hand, a major drawback of globally evaluated (N)MI is that no spatial information in the image is considered (Rueckert et al., 2000). We also empirically found that the training of NMI models are more dependent on spatial regularisation. Other similarity-invariant similarity metrics such as Normalised Gradient Fields (NGF) has also been explored in DL registration where rich amount of edges of the structures of interest can be found in the images (Hering and Heldmann, 2019), which could also be compared to for multi-modal registration.

Noticeably, folding are still present in experimental results for all methods despite using the diffeomorphic SVF transformation. If topology preservation is required for specific applications, it can be achieved by changing the hyperparameters. The velocity field could be enforced to be smoother to achieve diffeomorphism by increasing the smoothness regularisation, often at a cost of substantial drop in Dice accuracy; the number of time steps used in the Scaling-and-Squaring (SS) algorithm to approximate the continuous integral can be increased to reduce folding but with an increased computational cost. In this paper, we carefully tune the hyper-parameters within the constraints of our computational resource so that the presented results are more comparable for a general evaluation.

## 7. Conclusions

In this work, we present a deep learning framework trained using differentiable mutual information for fast and robust mono- and multi-modal image registration. We also propose to use a parameter-efficient B-spline free form deformation (FFD) via stationary velocity field (SVF) for smooth and diffeomorphic deformation. Evaluation results show that the proposed framework achieves competitive registration accuracy and transformation regularity across modality settings while being computationally more efficient. In future works, we will study the sensitivity of hyper-parameters, evaluate the different approaches on more diverse multi-resolution tasks, and investigate adding multi-resolution scheme and incorporating multi-step optimisation in DL registration.

### Acknowledgments

We thank EPSCRC grants EP/P001009/1 and EP/R005982/1 for supporting this work. Access to the cardiac MR data is granted under UK Biobank application 40119. We also thank Loic Le Folgoc, Jeremy Tan and Jo Schlemper from the BioMedIA group for discussions without which the work would have not been possible.

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

## Appendix A. Parzen window illustration

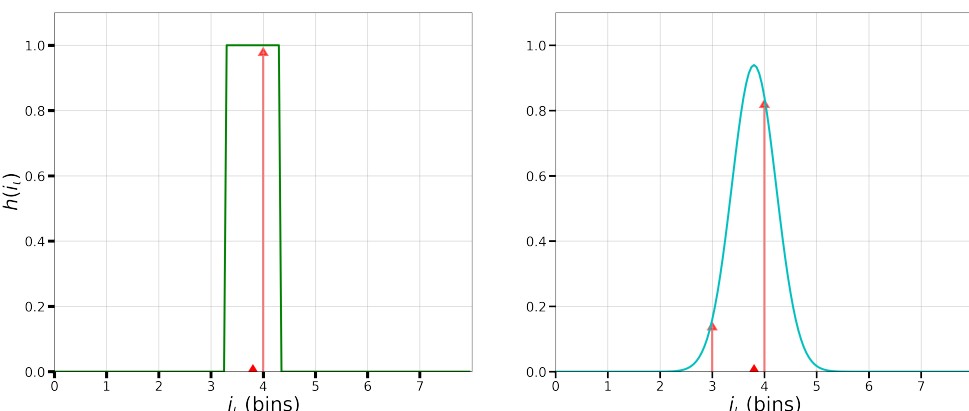

Figure 2: 1D illustration of histogram estimation using the rectangular window (left), known as *binning*, and Gaussian function as Parzen window (right). The horizontal axis shows the bin number in the intensity range. The red triangles mark the intensity value of one sample point in the image and the red arrows indicate the values that this sample contributes to the histogram at the evaluated bin centres $i_\iota$.

## Appendix B. Network architecture

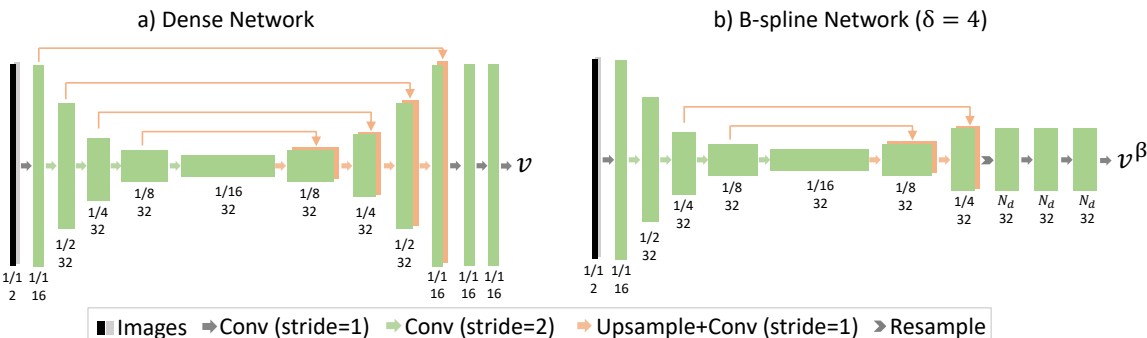

Figure 3: The architectures of: a) The base U-net architecture for direct prediction of dense fields; and b) Our network for prediction of the control point velocities $\boldsymbol{v}^\beta$ for B-spline parameterised SVFs. The example network shown here is configured for control point spacing of 4 (pixels/voxels). The resolution relative to the original images and number of channels are shown below each data block.

## Appendix C. Hyper-parameters

Table 2: Hyper-parameters of all methods producing the results shown in the paper. $\delta$ is the spacing of the B-spline control point grid (in image space). $\lambda$ is the regularisation weight introduced in 4, for SVFFD is the weighting on Bending Energy regularisation loss.

| Methods | Brain T1w-T1w | | Brain T1w-T2w | | Cardiac Motion | |
|---|---|---|---|---|---|---|
| | $\delta$ | $\lambda$ | $\delta$ | $\lambda$ | $\delta$ | $\lambda$ |
| SVFFD$_{LNCC}$ | 4 | $10^{-3}$ | 6 | $10^{-4}$ | 4 | $10^{-3}$ |
| VM$_{LNCC}$ | - | 0.1 | - | 0.1 | - | 0.1 |
| MIDIR$_{LNCC}$ | 2 | 0.1 | 2 | 0.1 | 2 | 0.1 |
| SVFFD$_{NMI}$ | 8 | $10^{-5}$ | 7 | $10^{-5}$ | 4 | $10^{-6}$ |
| VM$_{NMI}$ | - | 0.1 | - | 0.05 | - | 0.1 |
| MIDIR$_{NMI}$ | 2 | 0.08 | 2 | 0.1 | 2 | 0.1 |

Other hyper-parameters:

- Window size when computing LNCC: 7

- Number of *bins* used when computing NMI: 32 for SVFFD on cardiac registration, 64 for all other experiments

## Appendix D. Boxplots of quantitative results

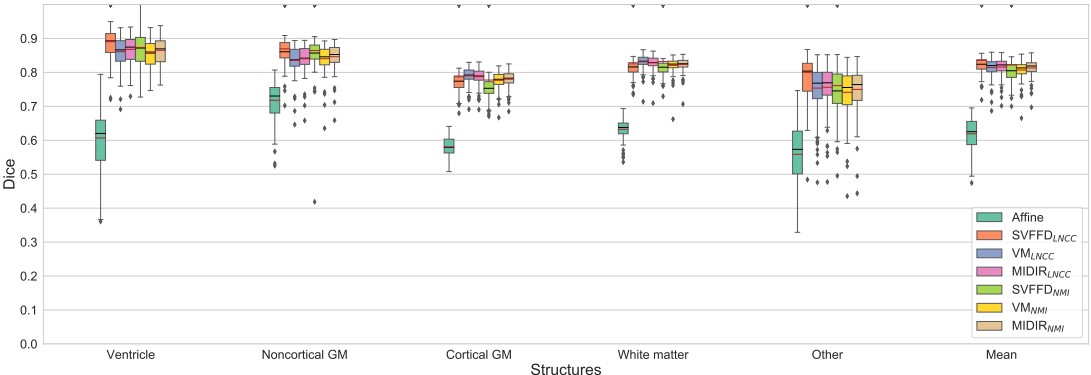

Figure 4: Boxplot of Dice results for brain T1w-T1w registration. The red lines in the boxes mark the mean value and the black lines mark the median. Results are shown for different groups of anatomical structures (GM stands for Grey Matter), with the Mean Dice over all structures on the right most.

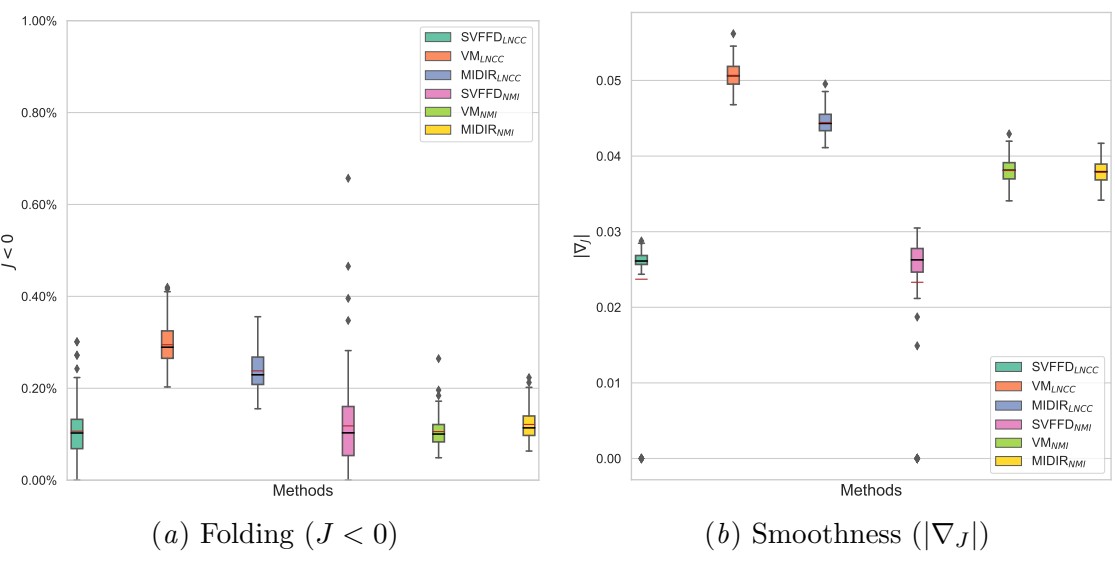

$(a)$ Folding $(J < 0)$        $(b)$ Smoothness $(|\nabla_J|)$

Figure 5: Boxplot of regularity results for brain T1w-T1w registration.

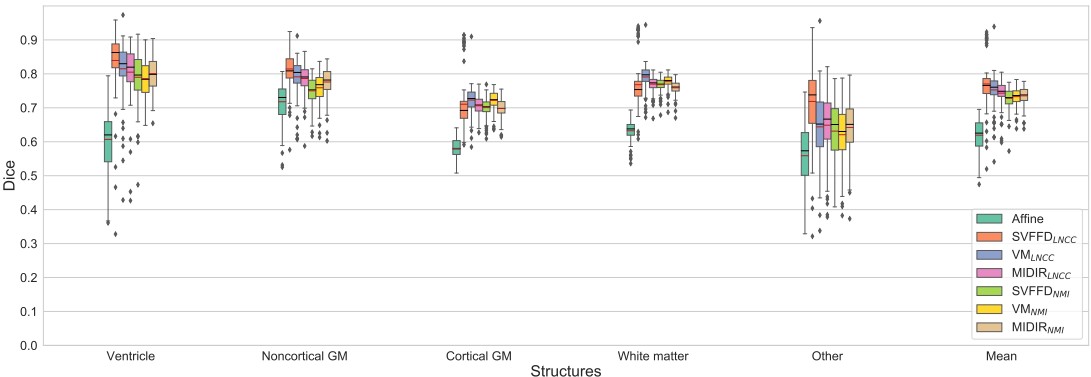

Figure 6: Boxplot of Dice results for brain T1w-T2w registration. Similar configuration as Figure 4.

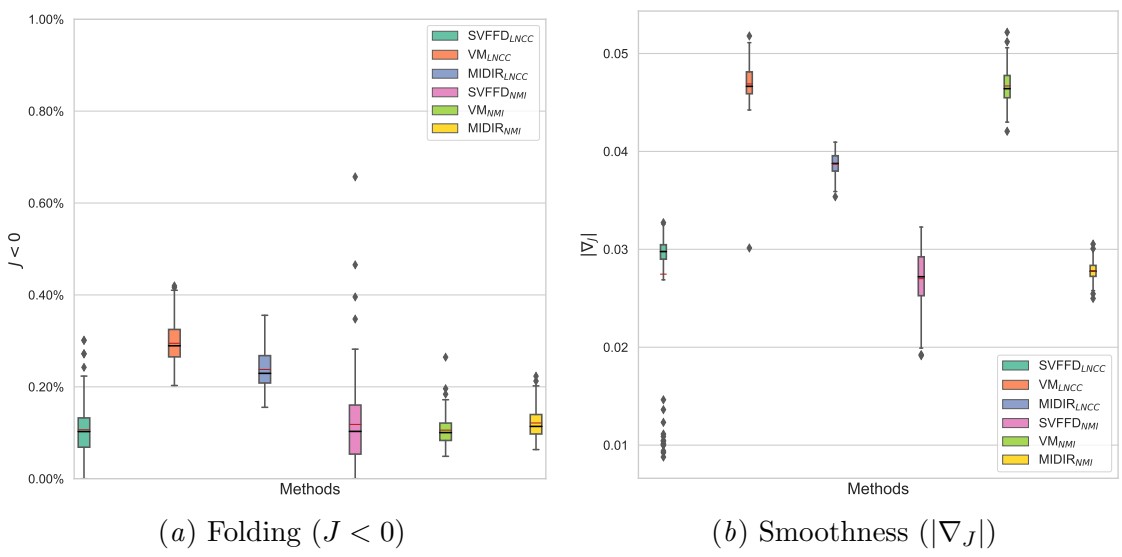

(a) Folding ($J < 0$)  (b) Smoothness ($|\nabla_J|$)

Figure 7: Boxplot of regularity results for brain T1w-T2w registration.

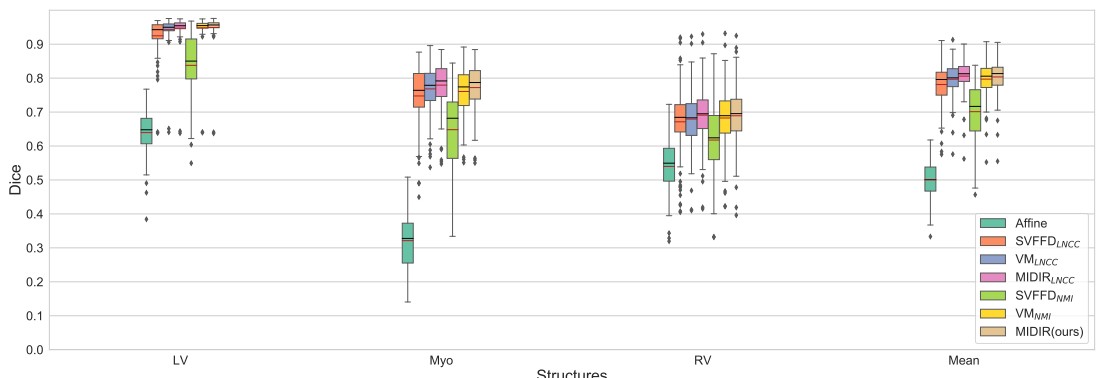

Figure 8: Boxplot of Dice results for cardiac MR registration. Results are shown for Left Ventricle volume (LV), Myocardium (Myo) and Right Ventricle volume (RV).

## Appendix E. Visualisation of images and transformations

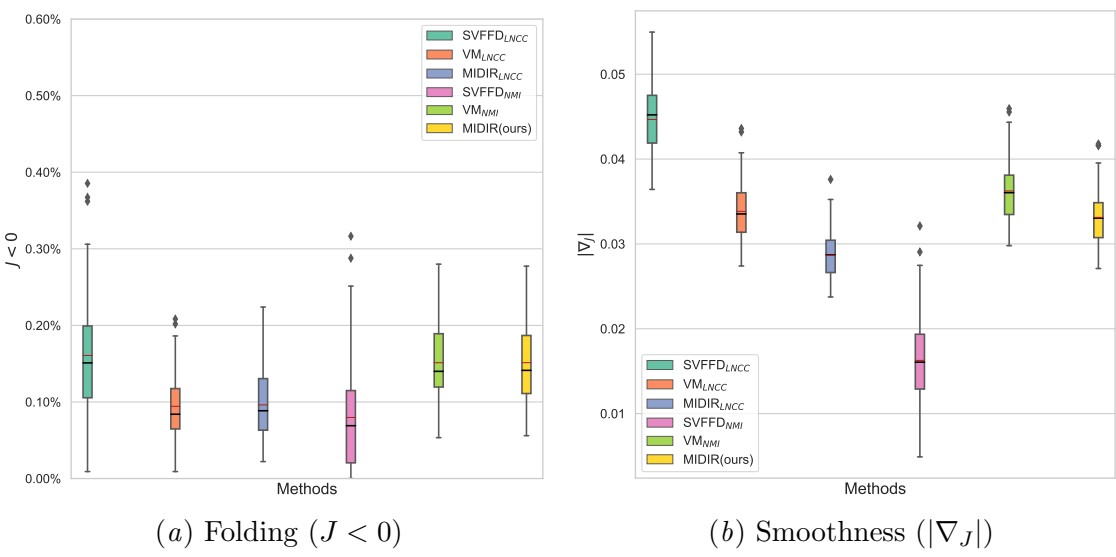

(a) Folding ($J < 0$)  (b) Smoothness ($|\nabla_J|$)

Figure 9: Boxplot of regularity results for cardiac MR registration.

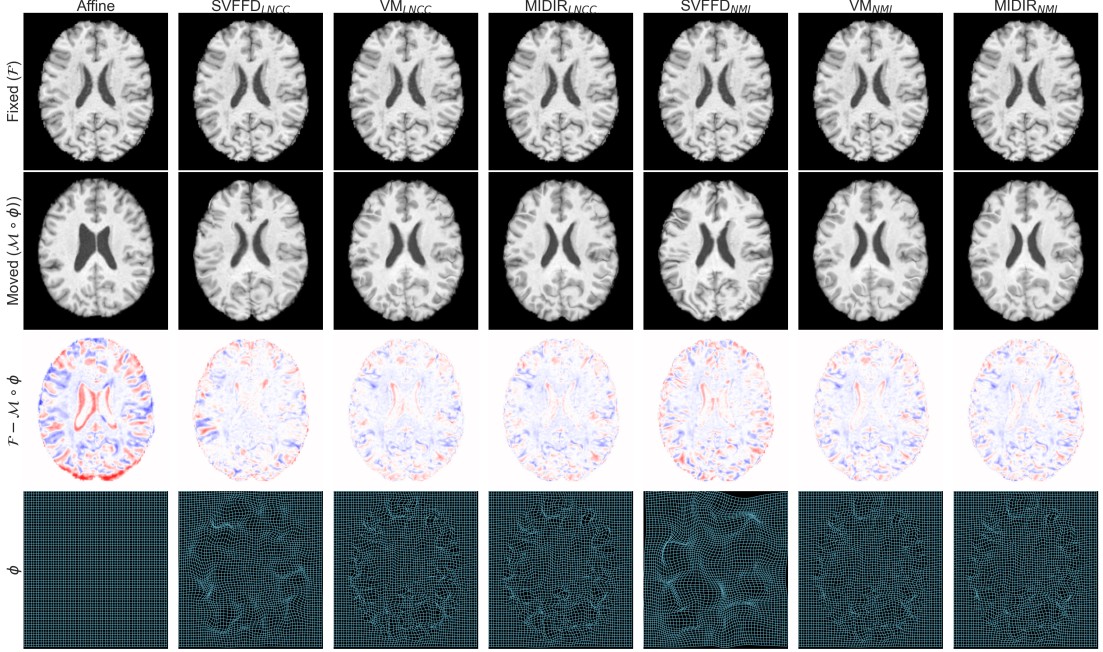

Figure 10: An example axial slice of brain MR T1w-T1w registration from all competing methods. The rows are the target fixed image, the moving image transformed by registration (moved), the error of the registration (white indicates zero error, red means positive and blue means negative) and the transformation.

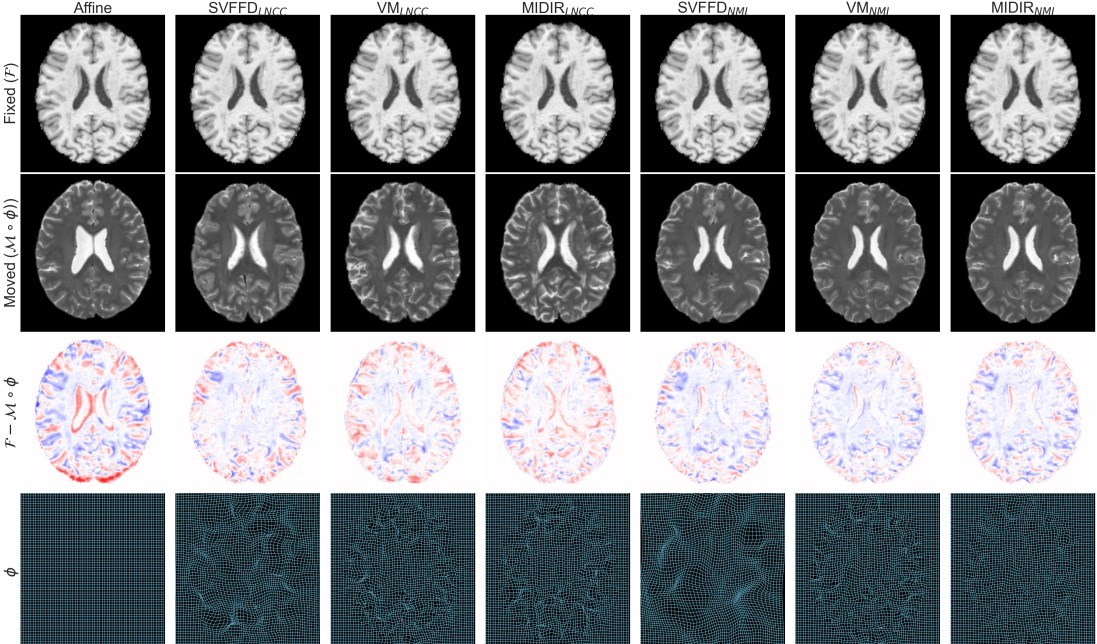

Figure 11: An example axial slice of brain MR T1w-T2w registration results. Same configuration as Figure 10, except the error is between the fixed image (T1w) and the transformed T1w image from the same subject as the moving T2w image, which is initially perfectly aligned with the T2w image.

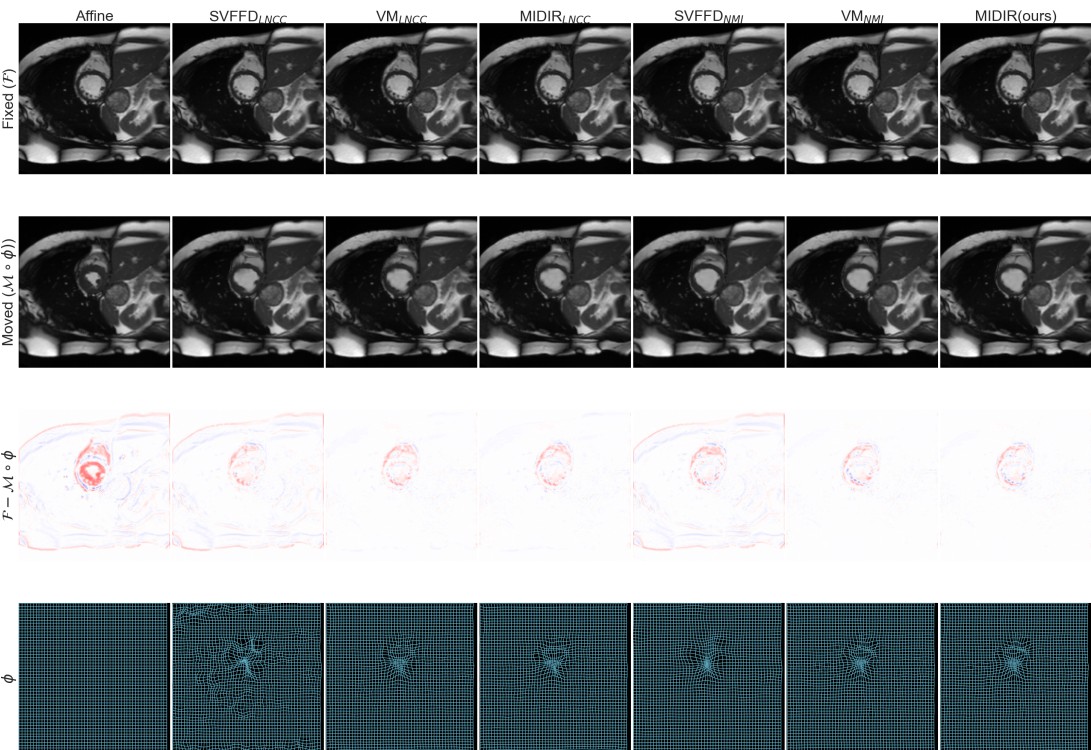

Figure 12: Example registration of an ED frame (fixed image) of mid-ventricle slice with the ES frame (moving image) of the same sequence. Same configuration as Figure 10.

