# OpenReview forum: "Learning Diffeomorphic and Modality-invariant Registration using B-splines"
_MIDL.io/2021/Conference — MIDL 2021_

### Official Review · AnonReviewer2 · 2021-03-05

**Confidence:** 2
**Preliminary Rating:** 4
**Recommendation:** Oral, Poster
**Final Rating:** 4

**Summary:**

The paper presents a deep learning framework for learning diffeomorphic and modality-invariant registration using B-splines. The main contribution of the proposed method is to use Parzen window-based mutual information estimation which allows gradient-based network training. Experiments on three different publicly available datasets show the effectiveness of the proposed method.

**Strengths:**

* Main strength of the proposed method is that it combines classical registration framework (B-Splines and MI) and adapts it to the deep learning framework.
* Really good literature review with good insights regarding their pros and cons.
* Experiments are thorough and comparison against state-of-the-art deep learning and non-deep-learning-based methods clearly identifies the advantages and disadvantages of the proposed method.

**Weaknesses:**

* A little bit more explanation about why the deep learning-based proposed framework gives inferior results compared to the non-deep-learning-based framework would be useful.
* Box plots given in the appendix are useful, but a clear statistical significance analysis in Table-1 is required.
* Showing the effect of varying different hyper-parameters (B-spline control point size, $\sigma$ in Parzen window, the parameter $\lambda$ in Eq.5) would make the paper stronger


**Deanonymize Review:**

no

**Detailed Comments:**

* Comparison against multi-modal image-to-image transformation-based registration framework could be helpful.


**Final Rating Justification:**

The authors answered the main questions raised during the review process.

**Justification Of The Preliminary Rating:**

The paper proposes a novel framework where deep learning is combined with classical registration formulation. This will result in a lot of insightful future discussion and can help the community start looking back at the (neglected) registration problem. Results are show the usefulness of the proposed method. Experiments are thorough. Promise to make code and hyper-parameters of the proposed method is an advantage.

**Paper Type:**

methodological development

**Questions To Address In The Rebuttal:**

Mainly, the first two points of the weakness section.

**Special Issue:**

yes

---

> ### Author Response · Authors · 2021-03-18
> **Response to AnonReviewer2**
>
> We thank Reviewer 2 for the positive notes on the contribution of our work and the quality of the literature review.
> 1. Performance of deep learning frameworks compared to classical registration:
> We agree that more discussion regarding this should be included. The classical registration method performs detailed optimisation for each pair of images, while the deep learning methods perform one-pass fast predictions but less accurately in its current form. A deep learning registration framework that effectively performs an iterative optimisation process for each pair of images at inference could be a better approach and is something that we are exploring. The classical method also employs a multi-resolution framework (3 levels of resolutions were used in our experiments), which helps with optimisation and registration accuracy. In contrast, the deep learning frameworks presented here only use one resolution. We are also working on adding a multi-resolution solution to our DL registration framework. We have added these discussions in the revised paper.
> ---
>
> 2. Significance testing:
> We thank Reviewer 2 for pointing this out. We have added statistical significance tests between the results of our method and baselines, using the Wilcoxon signed-rank tests. We found that the performance of our method is mostly on-par with the VoxelMorph (DL) baseline on monomodal tasks (T1w-T1w and cardiac), and achieves on-par Dice score accuracy but (statistically significant with p-value <<0.05) better regularity on the multimodal task (T1w-T2w). This analysis has been added to the revised paper.
> ---
> 3. Effect of varying hyper-parameter values:
> We appreciate Reviewer 2 for suggesting this. We agree that such studies will tell us more about the sensitivity of the method to the changes of hyper-parameters. Extensive experiments will be needed for this study and will be subject to our future works.

---

### Official Review · ~Alessa_Hering1 · 2021-03-06

**Confidence:** 5
**Preliminary Rating:** 2
**Recommendation:** Poster
**Final Rating:** 3

**Summary:**

The manuscript presents a deep-learning-based registration framework for multi-modal image registration using mutual information as a similarity measure. The authors combine several already existing building blocks into a new method. They evaluate their approach on three datasets (brain t1w-t1w, brain t1w-t2w,  and cardiac).

**Strengths:**

In general, I enjoyed reading the manuscript. It is well structured, has a clear motivation, and is well written.
- experiments were performed on three datasets (brain t1w-t1w, brain t1w-t2w,  and cardiac)
- a fair comparison to a conventional registration method was performed
- good evaluation checking the accuracy and plausibility of the registration results
- the hyperparameter tuning was performed on a held-out validation dataset
- clear and informative visualization of the results (unfortunately in the appendix due to page limit)


**Weaknesses:**

In general, I have the feeling that this paper reads a bit over positive due to different reasons.
1) The discussion is missing and therefore the weaknesses of the proposed method are not discussed.
2) The paper gives the impression that only MI is a suitable similarity measure for multimodal registration. However, NCC is also a multi-modal similarity measure. And also, the normalized gradient field (NGF) distance measure is not mentioned at all.
3) For me, the paper reads like the authors did something new. However, I would say that they combine already existing building blocks to a new method. (which is fine with me!). Nevertheless, this should be clear.

Furthermore:
-  it’s not clear to me why the (L)NCC version of the method was not evaluated on the t1w-t2w dataset since the NCC should be a proper similarity measure.
- the segmentation masks are used for the evaluation are automatically generated. To me, it is unclear how much of the registration accuracy differences might be introduced by the inaccurate segmentation.
- the authors claim to have a registration method producing diffeomorphic transformation (see title), however, there are still foldings inside the deformation field (0.121%, 0.023%, and 0.151%) which makes the transformation not diffeomorphic anymore. Is it because of the choice of hyperparameters? Again, it’s fine to me to not have a diffeomorphic transformation but it should be discussed.


**Deanonymize Review:**

yes

**Detailed Comments:**

-  p. 6 i.e. |\nabla | \nabla_\phi || - is that correct?
- Fig. 7 a) 5 methods b) 3 methods – shouldn’t it 3 for both? Or have you registered the t1w-t2w also with the other two methods and therefore you can/should report results for all 5?
- p. 7 “use normalized cross correlation (LNCC) as similarity.” LNCC -> locally normalized cross correlation


**Final Rating Justification:**

Thank you for this detailed response. I appreciate the made changes in the manuscript and think it is ready to be published.

Just a small remark:
In [1] the NGF distance measure is already used for CT-MR dl-registration and therefore, it might be the more suitable reference, but the chosen one is also fine.


[1] Hering, A., Kuckertz, S., Heldmann, S., & Heinrich, M. P. (2019). Memory-efficient 2.5 D convolutional transformer networks for multi-modal deformable registration with weak label supervision applied to whole-heart CT and MRI scans. International journal of computer assisted radiology and surgery, 14(11), 1901-1912.

**Justification Of The Preliminary Rating:**

In general, I like the paper. It has a clear motivation and it's well written. The experiments look solid and the results are well presented. However, I miss the discussion in this paper. There are some weaknesses of the presented work, but they aren't named or discussed at all. Moreover, a few references to other multi-modal registration approaches are missing in the related work/discussion as well.  I am looking forward to a revised version of this really interesting paper. And if these points will be addressed in the rebuttal, I am happy to accept the paper.

**Paper Type:**

methodological development

**Questions To Address In The Rebuttal:**

The strength and weaknesses section might read like the work is not good, however, I just like to have a fair discussion of existing work and about the weaknesses of your approach. It has not to be perfect yet, but you have to name and discuss weaknesses. I would like to see this work presented at MIDL, but first, a few points have to be addressed:

- add references to other dl-reg methods using NCC and NGF and discuss them. What are the advantages/drawbacks of MI?
- add a discussion to the paper
- Why are there still foldings inside the deformation field? Is there a way to completely prevent foldings (with other hyperparameters)? Does this affect the accuracy results? Please discuss it.
- Please add the experiments with LNCC as a similarity measure or alternatively with NGF.

(Typically, it is possible to exceed the page limit slightly in order to address required questions in the rebuttal - please ask the PC for it.)

**Special Issue:**

no

---

> ### Author Response · Authors · 2021-03-18
> **Response to AnonReviewer3**
>
> We are glad that Reviewer 3 enjoyed reading the manuscript and we really appreciate the very constructive comments. As a summary of our response to Reviewer 3, we have added experiments of LNCC on T1w-T2w registration, and a discussion section to discuss other multi-modal similarity metrics and the weaknesses/limitations of the method and the paper.
>
> 1. Discussion of (N)MI vs. (L)NCC and NGF:
> We thank Reviewer 3 for pointing out that we should mention and discuss other multi-modal similarity metrics such as LNCC and NGF. We do not intend to claim that MI is the only similarity metric that can handle the intensity changes in multi-modal registration, and we have clarified this in the revised paper. We have also added the references and  the discussion of LNCC and NGF and their uses in DL registration.
> ---
>
> 2. Remaining folding:
> We acknowledge that folding is still present in the experimental results that we are showing. There are two factors here that lead to folding in the Stationary Velocity Field transformation model. One is that the velocity field is not sufficiently smooth to achieve diffeomorphism. This can be reduced or eliminated by adding more regularisation (higher $\lambda$ in our framework) but at a cost of substantial drop in Dice accuracy. We observed this when tuning regularisation for both the traditional method and deep learning methods. The other factor is that the number of time steps used in the Scaling-and-Squaring (SS) algorithm to approximate the continuous integral is too small. Increasing the amount of time steps (reducing step size) can reduce folding but also increases computational costs. If less folding is required, as it is the case for some clinical applications where topology preservation is required, one can reduce/remove folding by changing these factors. For our paper, we make a conscious choice to carefully tune the hyper-parameters within the constraints of our computational resource so that the presented results are more comparable for a general evaluation.
> ---
>
> 3. Experiment with LNCC on T1w-T2w registration:
> As suggested by Reviewer 3, we added experiments testing LNCC as similarity on T1-T2 registration for both conventional baseline and deep learning methods and added the results to the paper. The results show that LNCC performs quite well on T1w-T2w registration, which is also reasonable given the intensity relationships between T1w and T2w images, when observed locally, is mostly an inversion. We appreciate Reviewer 3 for suggesting this more thorough test. And we will further investigate the general applicability of the similarity losses on other multi-modal tasks such as MR-CT or MR-PET registration where (L)NCC is less likely to work but (N)MI is expected to.
> ---
> We also appreciate Reviewer 3 for pointing out some detailed errors, which we have corrected in the revised paper. The discussions above are added in a more concise form since the page limit is still in effect for rebuttal stage. We will consider expanding them if it’s later allowed. We hope our revisions and responses have addressed the concerns of Reviewer 3 sufficiently.

---

> > ### Comment · AnonReviewer3 · 2021-03-21
> >
> > Thank you for this detailed response. I appreciate the made changes in the manuscript and think it is ready to be published.
> >
> > Just a small remark:
> > In [1] the NGF distance measure is already used for CT-MR dl-registration and therefore, it might be the more suitable reference, but the chosen one is also fine.

---

> > > ### Comment · AnonReviewer3 · 2021-03-22
> > > **Question**
> > >
> > > I was further thinking about the used Scaling-and-Squaring algorithm. What is the cost of one time step? How much longer does the training/inference take to have one more time step? And do you have an approximation of how many time steps you would need to obtain a deformation field without foldings?

---

### Official Review · AnonReviewer1 · 2021-03-08

**Confidence:** 4
**Preliminary Rating:** 4
**Recommendation:** Oral, Poster

**Summary:**

The major novelty of this paper is that the authors propose the use of a mutual information based loss for the deep learning based multi-modality registration. The resulting DL registration framework becomes thus  modality invariant. The MI is computed via a Parzen window formulation to enable a differentiable estimate for gradient-based network training. In addition, the paper proposes an efficient B-spline free-form deformation (FFD) parameterization via a stationary velocity field (SVF).

**Strengths:**

- overall very nice paper, well and clearly presented
- significant novelty, with new NMI based loss, Parzen window formulation
- output is velocities of B-spline parametrization (encoded in a grid) to reduce the number of estimated parameters, significantly more efficient to voxel-wise presentations,
- comparison to traditional B-spline free form deformation registration and voxel-based DL (both with traditional loss and with newly added NMI loss)
- NMI show significantly improved results for multi-modality, and B-spline FFD shows improved runtime and lower parametrization need


**Weaknesses:**

- the comparison is still quite limited as there is only comparison with 1 other method in the field
- There is no comparison/evaluation whether more efficient parametrization would result in potentially lower training size needs. Currently, there is little evidence why a b-spline FFD should be preferred.
- no significance testing across the testing conditions so unclear whether methods/settings are actually significantly different


**Deanonymize Review:**

no

**Justification Of The Preliminary Rating:**

The paper is nice contribution to the field of DL registration, extending it to multimodality. There is significant novelty towards an improved DL registration incorporating normalized mutual information, as well as an efficient B-spline based deformation encoding. Solid/nice work.

**Paper Type:**

methodological development

**Special Issue:**

yes

---

> ### Author Response · Authors · 2021-03-18
> **Response to AnonReviewer1**
>
> We thank Reviewer 1 for the positive feedback on the quality and novelty of the paper and constructive suggestions.
> 1. Comparison to other methods:
> We would like to point out that we have evaluated the method against two other methods that we think are the most relevant. One traditional iterative method (“SVFFD”)  with the same transformation model and similarity metrics, and one state-of-the-art deep learning registration framework (“VM”). We decided to focus our efforts by carefully evaluating the performance of these methods to ensure a fair comparison. However, we agree that further comparison with other methods can be useful and can be included in our future work.
> ---
> 2. Evaluation on efficient parameterisation and reasons for using B-spline FFD:
> Improved data efficiency follows parameterisation efficiency. This is indeed a potential benefit, and we appreciate Reviewer 1 for suggesting this. Using the B-spline FFD+SVF model enables faster inference, reduces the number of parameters in the network, and produces intrinsically smooth transformations. The small number of parameters also makes it easier to plug this module into other DL frameworks that require learning of deformations.
> ---
>
> 3. Significance testing:
> We thank Reviewer 1 for pointing this out. We have added statistical significance tests between the results of our method and baselines, using the Wilcoxon signed-rank test. We found that the performance of our method is mostly on-par with the VM (DL) baseline on monomodal tasks (T1w-T1w and cardiac), and achieves on-par Dice score accuracy but statistically significant (with p-value <<0.05) improved regularity on the multimodal task (T1w-T2w). This analysis has been added to the results session of the revised paper.

---

### Official Review · AnonReviewer4 · 2021-03-10

**Confidence:** 5
**Preliminary Rating:** 2

**Summary:**

This submission aims to handle the diffeomorphic image registration problem for different modalities. There are two main contributions: one is the differentiable mutual information for measuring image matching, and the other is a B-spline-based stationary velocity field for presenting the transformations. The method was applied to two tasks: 3D brain image registration between T1w and T2w, and 2D cardiac MR image registration.

**Strengths:**

The submission works on an interesting problem in image registration. Overall, the paper is well written and easy to understand. Although the proposed method is not brand-new, it looks reasonable and has the potential to solve the targeted research problem.

**Weaknesses:**

The main issue of this submission is the insufficient experiments and the weak performance. Firstly, it would be better to have a more clear presentation on the ablation study of the proposed method, which shows the effectiveness of each introduced component in the paper. Secondly, the improvement over VoxelMorph is so subtle that very likely the improvement is not significant. Thirdly, the Cardiac task is not a multimodal case, if I understand correctly. Lastly, the deformation in the images is not big enough to demonstrate the algorithm could successfully handle a large deformation and the matching of different appearances from multi-modality at the same time.

**Deanonymize Review:**

no

**Detailed Comments:**

Please give a detailed description of the parameter settings, like \sigma, \delta, C, etc.

The cubic B-spline presentation has a limited capacity, is it enough to cover the complexity of the velocity field in registering two 3D images?

**Justification Of The Preliminary Rating:**

Although the method looks reasonable, the novelty of the proposed method is incremental, and the experiments do not provide good support to the effectiveness of the method. More work should be done before publishing.

**Paper Type:**

methodological development

**Special Issue:**

no

---

> ### Author Response · Authors · 2021-03-18
> **Response to AnonReviewer4**
>
> We thank Reviewer 4 for recognising the quality of the manuscript and providing constructive feedback. Regarding the concerns raised by Reviewer 4:
>
> 1. Experiment presentation:
> We agree that the presentation of the results can be improved to provide a clearer comparison. Therefore, we have rearranged the entries in the result table and the boxplots, so that the results are firstly grouped by similarity loss to show the difference between registration frameworks (non-DL, VM with SVF, our DL with B-spline SVFFD). The difference made by the similarity metric can be reflected by comparing the corresponding framework entries between the two groups.
> ---
>
> 2. Significance of performance difference:
> We acknowledge the concern of Reviewer 4. To clarify the significance of the difference of performances, we have added statistical significance tests using the Wilcoxon signed-rank tests. We found that the performance difference between our method and VM in terms of Dice is not significant. The smoothness/regularity improvements is not significant on monomodal but is significant on the multimodal (T1w-T2w) task. We have also shown that our B-spline framework uses less parameters (~15% in the models reported) and is faster at inference.
>
> Overall, the focus of our paper is to provide mutual information as an alternative choice of a classic modality-robust similarity metric (that has been shown in literature useful for more challenging multi-modal tasks), and B-spline SVF as an intrinsically smooth, parameter-efficient transformation in deep learning registration. And we carefully evaluate these components in our experiments.
> ---
>
> 3. Cardiac task is not multimodal:
> We acknowledge that cardiac motion estimation is a mono-modal task. We choose to include this experiment to show the general applicability of our method on both mono-modal and multi-modal registration tasks, as well as on different anatomical structures.
> ---
>
> 4. Large deformation in multi-modal registration:
> We thank Reviewer 4 for the comment. The method of our paper does not focus on tackling this problem as we indicated in point 2). However, being able to handle large deformation is indeed important in some applications. And frameworks that use multi-resolution framework and/or multiple optimisation steps instead of one-step prediction could help with large deformation, and will be subjects of our future work.

---

### Author Response · Authors · 2021-03-18
**Comment to all reviewers summarising revision**

We thank all reviewers for their comments and constructive feedback. We have submitted a revised version of the paper. In addition to individual responses, we would also like to highlight the changes we made to the paper in the revision version:
- We have performed statistical significance tests to further evaluate the differences between methods, the result of which has been added to the Results section of the paper.
- We have added experiments and results using LNCC as a similarity metric on T1w-T2w registration task.
- We have added a separate Discussion section for some more in-depth discussion of the methods and experimental results.

---

### Meta-Review · Area_Chairs · 2021-03-29

**Recommendation:** Accept (Poster)

**Metareview:**

The authors present a machine-learning based diffeomorphic registration with the main contributions being an NMI loss function and B-spline representations.

The reviewers were initially split, with the main concerns being the lack of novelty and improved performance. NMI is by no means a novel loss function or the only one appropriate for modality-invariant registration -- in fact, it's even been used in DL-based registration in several papers by now or compared as baselines (just search multi-modal registration in the past 2 years). The results often missed statistical significance as noted by most reviewers, and when these were added most results of the present method did not yield improvements. The remaining main contribution seems to be the B-spline representation, which enables for an efficient representation compared to current implementations. The reviewers that scored the paper high seemed to appreciate this aspect.

Overall, I believe the paper is incremental, essentially combining existing developments (mostly the B-spline representation) into a DL framework that is well explained and executed. The average final reviewers leaned to accept. While borderline, I recommend acceptance, with the caveat that the authors should switch the emphasis to the B-spline implementation more than the multi-modal aspect, which has already been implemented in several DL papers, some which the authors included, and many more (as I said, a google scholar quick search will reveal quite a few papers).

**Paper Type:**

methodological development

---

### Decision · Program_Chairs · 2021-03-31

Accept